# Happiness and Cognitive Impairment Among Older Adults: Investigating the Mediational Roles of Disability, Depression, Social Contact Frequency, and Loneliness

**DOI:** 10.3390/ijerph16244954

**Published:** 2019-12-06

**Authors:** Jit Hui Tan, Edimansyah Abdin, Shazana Shahwan, Yunjue Zhang, Rajeswari Sambasivam, Janhavi Ajit Vaingankar, Rathi Mahendran, Hong Choon Chua, Siow Ann Chong, Mythily Subramaniam

**Affiliations:** 1Research Division, Institute of Mental Health, Singapore 539747, Singapore; jithui01@hotmail.com (J.H.T.); shazana_mohamed_shahwan@imh.com.sg (S.S.); yunjue_zhang@imh.com.sg (Y.Z.); rajeswari_sambasivam@imh.com.sg (R.S.); janhavi_vaingankar@imh.com.sg (J.A.V.); siow_ann_chong@imh.com.sg (S.A.C.); mythily@imh.com.sg (M.S.); 2Department of Psychological Medicine, National University of Singapore, Singapore 11907, Singapore; rathi_mahendran@nuhs.edu.sg; 3CEO Office, Institute of Mental Health, Singapore 539747, Singapore; hong_choon_chua@imh.com.sg

**Keywords:** happiness, cognitive impairment, older adults, social isolation, disability, depression

## Abstract

*Background:* Understanding the lower level of happiness among older adults with cognitive impairment has been a largely neglected issue. This study (1) reports on the level of happiness among older adults in Singapore and (2) examines the potential mediating roles of depression, disability, social contact frequency, and loneliness in the relationship between cognitive scores and happiness. *Methods:* Data for this study were extracted from the Well-being of the Singapore Elderly (WiSE) study: a cross-sectional; comprehensive single-phase survey conducted among Singapore citizens and permanent residents that were aged 60 years and above (*n* = 2565). The Geriatric Mental State examination (GMS) was administered to the participants. Questions pertaining to socio-demographic characteristics; happiness; loneliness; social contact; depression; and, disability were utilized in this study. Logistic regression analyses and mediation analyses were used to explore the correlates of happiness and potential mediating factors. *Results:* Overall, 96.2% of older adults in Singapore reported feeling either fairly happy or very happy. In the regression analysis, individuals of Malay descent, those who were married/cohabiting, or had higher education levels were more likely to report feeling happy. After controlling for socio-demographic factors, higher cognitive scores were associated with higher odds of reporting happiness. We found that the positive association between cognition and happiness was fully mediated by disability, depression, loneliness, and frequency of contact with friends. *Conclusion:* The majority of the older adult population reported feeling fairly or very happy. While cognitive impairment has shown limited reversibility in past studies, unhappiness among older adults with cognitive impairment might be potentially mitigated through interventions addressing accompanying issues of social isolation, disability, and depression

## 1. Introduction

Happiness, being defined as “the overall appreciation of one’s life-as-a-whole” [1], has been increasingly shown to confer a number of protective health benefits. Happier individuals tend to live longer lives, enjoy better physical health, and possess greater psychological resilience [1,2,3]. Lyyra and colleagues [4] showed that higher emotional well-being, of which happiness is a key marker, was associated with lower mortality even after controlling for physical, social, and cognitive functioning in a 10-year longitudinal study of octogenarians in Japan. With the demographic ageing being a concern in most developed nations, there has been increasing attention paid to successful ageing—which includes the maintenance of happiness and life satisfaction into old age [5,6]. Although recent studies have demonstrated relatively high levels of satisfaction and happiness among people of older ages, such positive outcomes are not the case for individuals with cognitive impairment [7,8,9,10]. For instance, Cooper, Bebbington, and Livingston [8] showed that poorer cognition was associated with lower happiness and higher likelihood of depression in older adults aged 65 and above in a comprehensive, cross-sectional study in seven low- and middle- income countries. Understanding the contributions to lower levels of happiness among older adults with cognitive impairment requires holistic consideration of multiple domains. Similar to the literature on quality of life, there is an emphasis on multiple facets of health and well-being associated with happiness [11]. Consequently, examining the physical, mental, and social factors that are commonly associated with cognitive impairment among older adults might help to inform the relationship between cognitive impairment and happiness. 

Within the social domain, cognitive impairment has been increasingly linked to social isolation [12,13,14,15]. A meta-analysis of longitudinal cohort studies investigating social relationships and incident dementia reported that lower social interaction—particularly less frequent social contact and participation—increases the risk of developing dementia, with effect sizes that were comparable to well-established risk factors, such as late-life depression [14]. Social relationships and participation, in turn, have consistently emerged as key predictors of happiness [8,16] social isolation might consequently play a crucial role in the relationship between cognitive impairment and happiness. In Cooper, Bebbington, and Livingston’s [8] aforementioned study, the authors also showed that the social network type mediated and moderated the relationship between cognitive impairment and happiness. They concluded that the type of social network is particularly crucial in buffering the negative impacts of cognitive impairment on happiness. While Cooper and colleagues focused on objective measures of social isolation, loneliness—the subjective evaluation of one’s social network and feelings of satisfaction or dissatisfaction regarding it—should also be considered [17]. Several studies have shown that, while objective social isolation and loneliness are moderately associated, they have differential impacts on incident dementia [18]; cognitive function [13,19]; and, happiness [16,17,20]. Consequently, objective social isolation and loneliness may both independently explain the relationship between cognitive impairment and happiness among older adults.

Aside from social factors, the psychological and physical domains may also mediate the relationship between cognitive impairment and happiness. Numerous studies have shown that individuals with depression often present greater cognitive impairment severity [21,22]. Depression also predicted incident mild cognitive impairment three years later among a Spanish sample of older adults [23]. Similarly, increasing functional disability has been posited as an indicator of persistent cognitive impairment, and the subsequent development of dementia [22,24,25]. As depression and functional disability are both associated with lower levels of life satisfaction and happiness, both of the variables may be potential mediators of the relationship between cognitive impairment and happiness [26].

The relationship between cognitive impairment and happiness among older adults is particularly crucial in Singapore—one of Asia’s fastest ageing countries. Singapore is a multi-ethnic and developed country in Southeast Asia with a resident population of 3.9 million, comprising Chinese (74.3%), Malays (13.4%), Indians (9.1%), and other ethnic groups (3.2%) [27]. Older adults t aged 60 years and above are expected to increase from 17.9% of the population to 30.7% by 2030 [6]. Although a 2008 survey among Singapore residents reported that most Singaporeans either felt quite happy or very happy about life, no recent study has comprehensively investigated happiness among older adults in Singapore [28]. As the prevalence of cognitive impairment and dementia increases in an ageing nation, it is increasingly pertinent to identify the pathways through which cognitive impairment influences happiness and, accordingly, the protective factors of happiness that can be addressed. 

In the current study, we report on the level and correlates of happiness among older adults in Singapore. We also examine the potential mediating roles of depression, disability, objective social isolation, and loneliness in the relationship between cognitive impairment and happiness.

## 2. Methods

The data that were used in this study were extracted from the Well-being of the Singapore Elderly (WiSE) study, a nationally representative and cross-sectional study of Singapore residents (citizens and permanent residents) aged 60 years and above, conducted between August 2012 and December 2013. A total of 2565 respondents completed the WiSE study, with a response rate of 65.6%. The participants were randomly selected from a national registry, including residents living in nursing homes or hospitals at the time of the survey. As stated, in Singapore, the Malays and Indians form the minority of the population. Hence, residents that were aged 75 years and older, and those of Malay and Indian ethnicity were oversampled to ensure a sufficient sample size for these population subgroups. All of the participants completed a written informed consent prior to commencing the questionnaire. Written consent was obtained from a legally accepted representative or next of kin in cases where the participants were unable to provide informed consent. The questionnaires were administered in English, Mandarin Chinese, Malay, Tamil, or in Chinese dialects: Cantonese, Hokkien, or Teochew. Professional survey interviewers conducted face-to-face interviews at respondents’ residences while using an online Computer Assisted Personal Interviewing application. Each interview took an average of two to three hours and it consisted of questions on socio-demographic background, health, cognition, and neurological tests. In addition, an informant was chosen for each selected respondent and it was also administered a questionnaire. An informant was defined as ‘the person who knows the older person best’ and the amount of time spent with the older person was used as the criterion for deciding the best informant in the event of multiple possible informants. The WiSE study obtained ethical approval from National Healthcare Group, Domain Specific Review Board, and the SingHealth Centralised Institutional Review Board. The article by Subramaniam et al. provides further details on the WiSE methodology [29]. 

## 3. Measures

### 3.1. Happiness

The primary outcome measure was a single question on happiness that was obtained from the Geriatric Mental State (GMS), a structured clinical mental state interview that applies the Automated Geriatric Examination for Computer Assisted Taxonomy (AGECAT) [30]. The participants were asked: “In general, how happy would you say you are: very happy, fairly happy, not very happy, or not very happy at all”. Single-item questions on happiness have been consistently used in studies across cultures [8,10,15] and have been tested for reliability and validity [31]. 

### 3.2. Cognitive Scores

The cognitive scores were obtained from a cognitive test battery comprising the Community Screening Instrument for Dementia (CSI-D) [32], which incorporated the Consortium to establish a Registry for Alzheimer’s Disease (CERAD) animal naming verbal fluency task, and the modified CERAD 10-word list learning task with delayed recall to generate a global cognitive score (COGSCORE) [33]. Each participant was assigned an item-weighted total score that was based on the number of correct answers.

### 3.3. Social Contact

Objective social isolation was assessed while using three contact frequency questions: (1) How often do you see any of your children or other relatives to speak to? (2) How often do you have a chat or do something with one of your friends? (3) How often do you see any of your neighbours to have a chat or do something with? Six response options were provided, ranging from “Daily” to “Never”. Higher scores indicated less frequent contact.

### 3.4. Loneliness

Loneliness was assessed while using a single question on the GMS-AGECAT: “Do you feel lonely?” The participants responded either “No”, “Yes but mild to moderate intensity, infrequent or fleeting”, or “Yes and severe, frequent or persistent”.

### 3.5. Depression

Depression diagnosis was assessed while using the GMS-AGECAT. The GMS-AGECAT, generates four syndrome clusters: organicity (dementia); schizophrenia and related paranoia; depression; and, anxiety neurosis. Five levels of psychopathology severity were generated for each syndrome, ranging from 0 (no symptoms) to 5 (very severely affected). Severity levels of 3 and greater constitute a ‘case’, while levels 1 and 2 represent ‘subcases’. The depression variable was dichotomized, with cases versus subcases. Reporting on concordance within a Singapore sample, Kua [34] showed that the AGECAT and psychiatrist’s diagnoses for depression achieved kappa values of 0.88. 

### 3.6. Disability

The World Health Organization Disability Assessment Schedule 2.0 was used to assess physical health and limitations (WHO-DAS II) [35]. The WHO-DAS II provides an overall disability score based on assessments of day-to-day functioning across six domains: cognition, mobility, self-care, getting along, life activities, and participation; item responses range from 0 (no difficulty) to 4 (extremely difficulty or cannot do). The total scores were used in the analyses.

### 3.7. Socio-Demographic Information

The socio-demographic questions included age at interview, ethnicity, gender, marital status, educational qualification, and employment status.

## 4. Statistical Analyses

Statistical analyses were conducted while using SAS system version 9.3. The data were weighted to adjust for oversampling and post-stratified by age and ethnicity between the survey sample and the Singapore resident population in 2010 to ensure that the sample was representative of the Singapore population. Descriptive analyses were first conducted to establish the level of happiness in the population. Prior to multivariate analyses, the happiness scores were transformed by collapsing the four categories (very happy, fairly happy, not very happy, not very happy at all) into two categories (not happy, happy), as there were very few cases in the ‘not very happy’ and ‘not very happy at all’ categories. Additionally, the loneliness scores were dichotomized into not lonely or lonely. Logistic regression analysis was performed to examine the socio-demographic correlates of happiness. Subsequently, we investigated the individual associations between happiness and each potential mediating factor (depression, WHO-DAS II score, loneliness, and contact frequency variables), while controlling for cognitive scores and socio-demographics. In this analysis, we hypothesized potential mediators, including depression, WHO-DAS II score, loneliness, and contact frequency variables, mediated the effects of cognitive impairment on happiness. We first conducted a series of logistic regression analyses by adding potential mediators that were significantly associated with happiness to test for mediation effects; in model 1, we added cognitive scores while controlling for socio-demographic factors; in model 2, we added depression and disability; and, in the 3rd. and final model, we added social contact variables and loneliness. Figure 1 presents a schematic diagram of the observed relationship between happiness, sociodemographic factors, mediators, and cognitive score in our mediation model. Following generally accepted criteria by Baron and Kenny [36], mediation would require that happiness was significantly associated with cognitive scores and the potential mediating factors; and, the effect size of the relationship between happiness and cognition would be reduced when potential mediating variables were added. Given the outcome is a binary variable, the decomposition of total effect into direct and indirect effects in a logit model was conducted using Karlson Holm Breen model [37]. A direct effect refers to the effect of cognitive scores on happiness without the mediating effects, while the indirect effects refer to the effect of cognitive scores on happiness via the mediator. The total effect refers to the combination of both direct and indirect effects of cognitive scores on happiness. Statistical significance was set at *p* < 0.05, while using two-sided tests.

## 5. Results

### 5.1. Level of Happiness

Out of the 2565 respondents that were interviewed in the WiSE study, 2374 individuals responded to the question on happiness and were included in the analyses. Overall, 96.2 % (*n* = 2269) reported that they were either very happy (*n* = 748, 35.8%) or fairly happy (*n =* 1521, 60.4%). Only 3.8% (*n* = 105) of respondents reported being either not very happy (*n* = 84, 3.2%) or not very happy at all (*n* = 21, 0.6%). Table 1 presents the socio-demographic characteristics and the health-related factors of our sample. 

### 5.2. Socio-Demographic Correlates of Happiness

Table 2 presents the socio-demographic correlates of happiness in this sample. The odds of happiness were significantly lower among Indians (OR: 0.55, 95% CI: 0.33–0.91; *p* = 0.02) and Other ethnicities (OR: 0.25, 95% CI: 0.06–0.99; *p* = 0.048) when compared to those of Chinese ethnicity. Additionally, individuals who were divorced or separated were significantly less likely to report being very/fairly happy than those who were married or cohabiting (OR: 0.23, 95% CI: 0.09–0.59; *p* = 0.002). Age, gender, employment status, and educational attainment were not significantly associated with happiness. 

### 5.3. Happiness and Cognitive Impairment 

Higher cognitive scores were associated with higher odds of reporting happiness after controlling for socio-demographic factors into the logistic regression model, (OR: 1.11, 95% CI: 1.01–1.22; *p* = 0.02) (Table is available upon request). After adding depression, WHO-DAS II score, loneliness, and the significant contact frequency variables (contact with friends and contact with neighbours) in the regression model, we found individuals with depression (OR: 0.23, 95% CI: 0.09–0.58; *p* = 0.002) or greater disability (OR: 0.97, 95% CI: 0.95–0.99; *p* = 0.009) had lower odds of being happy. Individuals who were lonely (OR: 0.39, 95% CI: 0.18–0.81, *p* = 0.01) or who had less frequent contact with friends (OR: 0.82, 95% CI: 0.68–0.10; *p* = 0.046) also had lower odds of being happy. However, cognitive score was no longer significantly associated with happiness in this regression model (Table 3), suggesting that it was fully mediated by the other variables.

## 6. Mediational Effects

Hence, potential mediators, including depression, WHO-DAS II, loneliness, and contact with friends, were tested. Table 4 summarizes the mediational effects. A significant indirect effect was found for all potential mediators tested after including all of the mediators in mediation analysis. The WHO-DAS II explained the largest percentage of the association (96.89%), followed by depression (13.06%), loneliness (12.32%), and contact with friends (8.67%).

## 7. Discussion

In a representative sample of elderly people in Singapore, 96.2% reported feeling either fairly or very happy. This level supports the past findings of relatively high levels of happiness among older adults [5,8,10]. Some authors have attributed this happiness to the greater emotional regulation exercised in later life, in which older adults are more selective of the social situations and the individuals whom they interact with [38]. Additionally, we found that individuals with lower cognitive scores were less likely to report feeling happy. This corroborates Cooper, Bebbington, and Livingston’s [8] findings of better cognition being associated with happiness across multiple low and middle income countries.

We sought to understand the negative association between cognition and happiness by exploring the physical, mental, and social factors that may mediate the relationship. Disability, depression, loneliness, and frequency of contact with friends emerged as significant mediators of the relationship between the cognitive scores and happiness, with disability and depression demonstrating the two largest mediation effects. Hirosaki et al. [26] had similarly shown that lower disability and the lack of depressive symptoms were two key predictors of self-rated happiness among a sample of community-dwelling Japanese elderly. Additionally, cognitive functioning was not associated with happiness when functional ability, depressive symptoms, and other socio-demographic factors were controlled for. Cooper et al. [7] had also reported that cognitive impairment itself does not explain the lower quality of life among individuals with cognitive impairment; instead, physical and mental disability fully mediated the relationship between cognitive impairment and a lower quality of life. Conversely, Cooper, Bebbington, and Livingston [8] had found that depression was not a mediator of the relationship between cognitive impairment and happiness; the authors showed that, even after controlling for WHO-DAS and depression, cognitive impairment was still significantly associated with a lower level of happiness. These contradicting findings to our study may be due to the differences in sample characteristics; Cooper and colleagues’ [8] data had been collected from multiple low and middle income countries of varying cultures, whereas our data is specific to a high income Asian country. Further research is needed to better understand cross-cultural and economic differences in how individuals respond to cognitive impairment and its impact on happiness. Disability has been shown to have a debilitating influence on happiness and life satisfaction, while a positive affect has demonstrated the protective effects on disability among elderly people [24,39,40,41]. Depression and disability have both been posited as close accompaniments to cognitive impairment, with some studies even citing depression as a risk factor for cognitive impairment [22,23,24]. Our findings suggest that happiness is diminished due to the physical and mental disabilities that accompany cognitive impairment, rather than by cognitive impairment per se. Thus, interventions targeted to reduce disability and depression may prove to be beneficial to older adults’ subjective well-being. 

Past research has focused on objective forms of social isolation, such as frequency of contact, but recent authors have posited the importance of studying both subjective and objective forms, respectively [18,19]. We sought to fill this gap in the literature by investigating both loneliness and objective social isolation as potential mediators. Our study showed that loneliness and the frequency of contact with friends were significant mediators of the relationship between happiness and cognitive impairment, in which loneliness was a larger mediator than objective social isolation. Holwerda and colleagues [18] showed that loneliness was associated with a greater risk of dementia three years later, but not social isolation. Additionally, there was a greater decline in the cognitive scores among individuals who reported higher levels of loneliness at the baseline [18]. These findings suggest that the unhappiness associated with cognitive impairment might be attributable to greater subjective feelings of loneliness, irrespective of tangible social contacts, while various forms of social isolation may have a negative impact on happiness [16,17,20]. Interestingly, only contact with friends was a mediator, while contact with children and family was not significantly associated with happiness. Additionally, we attempted to further understand the mediating effects in post-hoc analyses by including interaction terms between cognitive scores and mediating variables, e.g. cognitive scores x depression, into the mediation model. Only the interaction between cognitive scores and contact with friends was a significant mediator. This suggests that, for individuals with higher levels of cognitive scores (or lower cognitive impairment), having more contact with friends significantly influences their levels of happiness. Studies have shown that people with cognitive impairment tend to have smaller social circles, which may be a result of a gradual loss in social interaction skills [18,42]. A meta-analysis of longitudinal studies investigating social relationships and the risk of dementia found that it was the lack of the social interaction element that was the most predictive of incident dementia [14]. Additionally, Cooper et al. [8] found that cognitively impaired individuals with family dependent social networks were the least likely to report feeling happy. These findings seem to suggest that cognitive impairment has more detrimental effects on social networks with friends than with relatives. It could also be that an individual’s happiness is more vulnerable to changes within their social sphere with friends, whereas interaction with family members might represent a more stable and consistent aspect of the social network that minimally influences one’s evaluation of happiness.

With regards to socio-demographic outcomes, Indians and Other ethnic groups initially showed lower likelihoods of being happy compared to Chinese people. However, this relationship disappeared after controlling for the mediating factors. Instead, the adjusted analyses showed that Malays have higher odds of being happy compared to Chinese. Conversely, even after health and social factors are held constant, Malays seem to have additional factors that contribute to the more positive affect than the Chinese. Interestingly, a previous study that was conducted on the quality of life in Singapore found that significantly more Muslims reported feeling very happy about life as compared to the other religions [28]. Perhaps variables that were not measured, such as religion or community values within the Malay-Muslim community, play important roles in evaluations of well-being [43,44]. Moreover, we were unable to explore cultural-specific perceptions of happiness, which have been shown to influence reports of happiness [45]. Further empirical analyses are necessary to understand culturally intrinsic factors that promote happiness, as no previous study has investigated the predictors or interpretations of happiness across the diverse ethnic groups in Singapore. Our findings also reinforce the protective effects of marriage and higher education on happiness and satisfaction [8,15,39]. Individuals that were never married were less likely to be happy than those who were married or cohabiting; and, the incompletion of primary school education was associated with a lower likelihood of being happy than if one were to complete a tertiary education.

## 8. Limitations

The limitations of this study should be taken into account when interpreting the findings. First, the happiness scores were highly skewed with limited range in unhappy responses. We had to dichotomize the happiness variable due to limited cases in the ‘not very happy’ and ‘not very happy at all’ responses. While this was interpreted as a high level of happiness among older adults in Singapore, the low level of unhappy respondents and its restricted variance may have also limited our analyses in identifying the significant predictors of unhappiness. Additionally, happiness and loneliness were both assessed while using single items. While past studies have generally relied on single-item measures of happiness and loneliness, both variables are abstract constructs that may be more fully captured by using multiple items [5,10,15]. Second, the current study was based on secondary analysis of data from the Wellbeing of Singapore Elderly Study that was undertaken in 2012/2013. After excluding respondents with 10/66 dementia, only 5.1% of our older adult sample met criteria for cognitive impairment. We followed precedence by operationalizing cognitive impairment as two standard deviations below the mean, due to the lack of normative data on the cognitive profile of Singaporean older adults [46]. Such low prevalence might again limit our statistical power in detecting the influence of low cognitive scores and its interaction effects on happiness. Third, as this is a cross-sectional study, it is not possible to establish causal or temporal relationships between cognitive impairment, happiness, and the significant mediators. Fourth, we only operationalized objective social isolation through the frequency of contact; however, there are several other measures, such as social network size, living arrangements, or social participation, which we did not measure. Perhaps, these other forms of social isolation may play a more significant role in the relationship between cognitive impairment and happiness than what we have reported. Lastly, the generalizability of the findings should be done with caution. Our sample was specific to the demographic of Singapore, which might considerably differ from other populations in terms of cultural norms, ethnic distribution, or political and economic climate.

## 9. Conclusions

This study was the first to establish the high level of happiness across a representative and large sample of older adults in Singapore. Additionally, an understanding of the association of happiness with cognition has been limited, particularly amongst a multi-ethnic Asian population. The current study addressed this gap by demonstrating that disability, depression, loneliness, and infrequent contact with friends are independent pathways through which cognitive impairment has a negative impact on happiness among older adults. Although cognitive impairment and dementia among the elderly have shown limited reversibility, disability, depression, and social isolation are issues that can be mitigated [22]. Future studies should look into the effectiveness of increasing levels of happiness and well-being through specific interventions on these mediators. As populations age and cognitive impairments become more prevalent, it is necessary to look beyond just eradicating illnesses, and instead look towards enabling a positive ageing journey.

## Figures and Tables

**Figure 1 ijerph-16-04954-f001:**
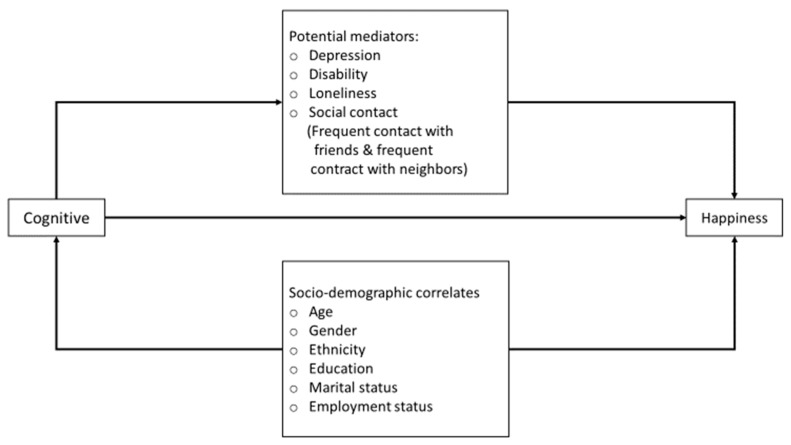
A schematic diagram of the observed relationship between cognitive score, sociodemographic factors, mediators and happiness.

**Table 1 ijerph-16-04954-t001:** Socio-demographic characteristics and health-related factors of the sample.

Variable	Category	*N*	Unweighted %	Weighted %
Age group	60–74 years	1471	61.96	77.05
	75–84 years	618	26.03	18.83
	85 years and above	285	12.01	4.12
Gender	Male	1055	44.44	44.07
	Female	1319	55.56	55.93
Marital status	Never married	127	5.35	7.9
	Married/cohabiting	1430	60.29	65.16
	Widowed	710	29.93	21.31
	Divorced/separated	105	4.43	5.63
Education	None	421	17.79	15.53
	Some, but did not complete primary	579	24.46	23.86
	Completed primary	614	25.94	25.05
	Completed secondary	500	21.12	22.84
	Completed tertiary	253	10.69	12.72
Employment status	Paid work (part-time and full-time)	686	29.15	35.08
	Unemployed (looking for job)	32	1.36	1.6
	Homemaker	732	31.11	26.25
	Retired	903	38.38	37.07
Ethnicity	Chinese	935	39.39	83.25
	Malay	687	28.94	9.31
	Indian	717	30.20	5.95
	Other	35	1.47	1.48
**Depression**		164	3.7	0.5
**Loneliness**		403	12.5	0.9
			Mean	SE
**Cognitive score**			28.9	0.07
**WHO-DAS II**			8.4	0.4
**Frequency of contact with friends**			3.4	0.6
**Frequency of contact with neighbours**			3.5	0.05

*Note.* OR in bold represent those with *p*-values < 0.05.

**Table 2 ijerph-16-04954-t002:** Socio-demographic correlates of happiness.

	Happiness			
	95% CI			
	Odds Ratio	Lower Limit	Upper Limit	*p*-Value
Age group				
60–74	Ref.			
75–84	1.02	0.47	2.19	0.97
85 years and aboveGender	0.61	0.26	1.47	0.27
Male	Ref.			
Female	0.79	0.37	1.68	0.54
Marital status				
Married/cohabiting	Ref.			
Divorced/separated	0.23	0.09	0.59	0.002 *
Never married	0.41	0.15	1.14	0.09
Widowed	1.11	0.55	2.24	0.77
Education				
Completed tertiary	Ref.			
Completed primary	0.54	0.14	2.08	0.37
Completed secondary	1.07	0.27	4.23	0.91
None	0.41	0.10	1.67	0.21
Some, but did not complete primary	0.32	0.09	1.14	0.08
Employment status				
Paid work (part time and full time)	Ref.			
Homemaker	1.01	0.38	2.73	0.98
Retired	0.66	0.30	1.44	0.30
Unemployed	0.26	0.06	1.06	0.06
Ethnicity				
Chinese	Ref.			
Indian	0.55	0.33	0.91	0.02 *
Malay	1.42	0.77	2.63	0.27
Other	0.25	0.06	0.99	0.048 *

* *p*-value < 0.05.

**Table 3 ijerph-16-04954-t003:** Multiple logistic regression model with happiness as the outcome variable.

Criterion Variable	95% CI			
	Odds Ratio	Lower Limit	Upper Limit	*p*-Value
Cognitive score	0.94	0.82	1.07	0.34
Depression	0.23	0.09	0.58	0.002 *
WHO-DAS II	0.97	0.95	0.99	0.009 *
Loneliness	0.39	0.18	0.81	0.012 *
Frequency of contact with friends	0.82	0.68	0.10	0.046 *
Frequency of contact with neighbours	0.93	0.78	1.11	0.43
Age group				
60–74	Ref.			
75–84	1.69	0.72	3.98	0.23
85 and above	1.36	0.49	3.82	0.56
Gender				
Male	Ref.			
Female	0.95	0.45	1.99	0.89
Marital status				
Married/Cohabiting	Ref.			
Divorced/Separated	0.25	0.09	0.67	0.006 *
Never married	0.39	0.13	1.19	0.10
Widowed	1.18	0.56	2.48	0.67
Education				
Completed tertiary	Ref.			
Completed primary	0.51	0.13	2.05	0.34
Completed secondary	0.82	0.20	3.33	0.78
None	0.37	0.08	1.67	0.20
Some, but did not complete primary	0.24	0.06	0.90	0.03 *
Employment status				
Paid work (part time and full time)	Ref.			
Homemaker	1.17	0.36	3.79	0.79
Retired	0.78	0.33	1.84	0.57
Unemployed	0.30	0.07	1.23	0.09
Ethnicity				
Chinese	Ref.			
Malay	2.52	1.17	5.40	0.018 *
Indian	0.81	0.45	1.47	0.49
Other	0.25	0.06	1.00	0.05

* *p*-value < 0.05.

**Table 4 ijerph-16-04954-t004:** Mediational effects in the relationship between happiness and cognitive score.

Mediator	Effect	OR (95% CI)	*p*-Value	% Mediated	Adjusted % Mediated
Depression	Total	1.09	0.02	9.97	13.06
	Direct	1.07	0.07		
	Indirect	1.02	0.002		
WHO-DAS II	Total	1.07	0.07	74.00	96.89
	Direct	0.96	0.28		
	Indirect	1.12	0.00		
Loneliness	Total	1.10	0.02	9.41	12.32
	Direct	1.08	0.045		
	Indirect	1.01	0.02		
Contact with friends	Total	1.09	0.02	6.62	8.67
	Direct	1.08	0.045		
	Indirect	1.01	0.02		

*Note*. Models are adjusted for socio-demographic characteristics.

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
