# Peer review of "Happiness and Cognitive Impairment Among Older Adults: Investigating the Mediational Roles of Disability, Depression, Social Contact Frequency, and Loneliness"

_ijerph, 2019, doi:10.3390/ijerph16244954_

Round 1

Reviewer 1 Report

The manuscript is a well written report of a longitudinal study regarding a little more than 2500 older persons in Singapore aimed at relating the decline in cognitive function with the individual perception of happiness. Data for the study were extracted from the Well-being of the Singapore Elderly (WiSE) study and, apart from the evaluation of cognitive impairment with appropriate geriatric questionnaire, logistic regression analyses and mediation analyses were used to establish correlates of happiness and
potential mediatiors. Findings are resumed in complete and clear-cut tables and discussion is straightforward, however a graphical representation of at least a selected set of correlates (for ex.  a schematic diagram of the bivariate analyses; the individual associations between happiness and the mediational impact on it; etc) should be added in order to give a visual and immediate take home message to the reader. 

Author Response

Response to reviewer 1 comments

Comments and Suggestions for Authors

The manuscript is a well written report of a longitudinal study regarding a little more than 2500 older persons in Singapore aimed at relating the decline in cognitive function with the individual perception of happiness. Data for the study were extracted from the Well-being of the Singapore Elderly (WiSE) study and, apart from the evaluation of cognitive impairment with appropriate geriatric questionnaire, logistic regression analyses and mediation analyses were used to establish correlates of happiness and potential mediatiors. Findings are resumed in complete and clear-cut tables and discussion is straightforward, however a graphical representation of at least a selected set of correlates (for ex.  a schematic diagram of the bivariate analyses; the individual associations between happiness and the mediational impact on it; etc) should be added in order to give a visual and immediate take home message to the reader. 

Authors’ reply: We thank the reviewer for the comments. We have added a schematic diagram of the observed relationship between happiness, sociodemographic factors, mediators and cognitive score in the revised manuscript.

Reviewer 2 Report

The manuscript presents an intriguing study that examines the correlations between happiness and socio-demographic variables and certain health variables such as cognition, depression and disability.

The study is a secondary analysis of a cross-sectional study conducted on a representative sample of older people in Singapore. There is no strong hypothesis described in the introduction, and I was wondering if the results were not the result of multiple statistical series. It would have been preferable to use statistics to confirm or disprove a predefined hypothesis.

The main result of the study is based on the analysis of a single simple question about how participants felt happy. Participants were able to answer on the four-level scale. This four-level scale is rather imperfect since the last proposal is "not very happy at all" ("not happy at all" would have been better). The scale therefore discourages people from declaring themselves unhappy. The authors transform the results into a two-level variable for analysis. However, by doing this, almost all participants are considered "happy". This limitation was highlighted by the authors during the discussion. Examining the responses at four levels could provide an opportunity to study a gradient effect.

The values of variables related to depression, disability, loneliness and social contact are not given, and the variables that appear are only correlated with the level of happiness.

It is not clear that the feeling of happiness can be interesting to understand the health and functioning of older people. The answer probably reflects a generally positive emotional feeling, but it does not help to understand why the person is happy or not. The prevalence of happiness is a surprising term, since happiness is not a disease. Perhaps the study could find a better place in the sociology journal rather than in a medical journal.

Author Response

Response to reviewer 2 comments

Comments and Suggestions for Authors

The manuscript presents an intriguing study that examines the correlations between happiness and socio-demographic variables and certain health variables such as cognition, depression and disability.

The study is a secondary analysis of a cross-sectional study conducted on a representative sample of older people in Singapore. There is no strong hypothesis described in the introduction, and I was wondering if the results were not the result of multiple statistical series. It would have been preferable to use statistics to confirm or disprove a predefined hypothesis.

Authors’ reply: The results were derived from the multivariate analysis to test our hypothesized mediation model. We have added a schematic diagram of the observed relationship between happiness, sociodemographic factors, mediators and cognitive score in the revised manuscript for ease reference.

The main result of the study is based on the analysis of a single simple question about how participants felt happy. Participants were able to answer on the four-level scale. This four-level scale is rather imperfect since the last proposal is "not very happy at all" ("not happy at all" would have been better). The scale therefore discourages people from declaring themselves unhappy. The authors transform the results into a two-level variable for analysis. However, by doing this, almost all participants are considered "happy". This limitation was highlighted by the authors during the discussion. Examining the responses at four levels could provide an opportunity to study a gradient effect.

Authors’ reply: We would like to respectfully disagree with the reviewer’s opinion. As stated earlier in the manuscript, we were unable to examine the four levels results because the data is very skewed which results in the regression model becoming unstable. Hence, we have decided to keep the current analysis as it is.

The values of variables related to depression, disability, loneliness, and social contact are not given, and the variables that appear are only correlated with the level of happiness.

Authors’ reply: We have included the descriptive values of depression, disability, loneliness and social contact in the revised Table 1.

 It is not clear that the feeling of happiness can be interesting to understand the health and functioning of older people. The answer probably reflects a generally positive emotional feeling, but it does not help to understand why the person is happy or not.

Authors’ reply: We agree with the reviewer’s comments. We recognize that it could reflect both positive affect and/or an overall evaluation of their contentment in life. However, the scope of this paper does not examine the reasons behind their happiness but rather the associations between their ratings and other outcomes. Hence we were unable to provide on this aspect in this study.

The prevalence of happiness is a surprising term since happiness is not a disease. Perhaps the study could find a better place in the sociology journal rather than in a medical journal.

Authors’ reply: We agree with the reviewer’s comments. To avoid confusion in the interpretation of the results, we have replaced the term ‘prevalence’ with ‘level’ throughout the paper.

Reviewer 3 Report

This is a well-written paper that sets out a clear area for investigation. The methodology is cogently described and this provides a good platform for the findings. The discussion builds effectively on the findings and relates insights from this study to the literature corpus. My comments are limited to some suggestions and points of detail.

The first citation (Veenhoven, 2008) could be placed directly after the quote rather than at the end of the sentence (as a direct quote is utilised). 

It is mentioned in the methods that people of Malay and Indian ethnicity were oversampled. Does this require a degree of additional explanation? As a reader unfamiliar with the population of Singapore, I'm left unsure as to why this was necessary. Is it because people from these groups are harder to recruit to research?

It is clear that secondary analysis has been undertaken. However, in the discussion of the survey, in the methods section, the writing slightly lapses into a sense that this paper is based on primary research. A minor rewrite can underscore that this methodological discussion is referring to the previously collected data.

The data collection was undertaken in 2012/13 - is it worth acknowledging this point in the limitations section?

It is a minor point, but data can be treated as plural in a scientific paper - so in the abstract under the methods heading, it can be presented as 'data from this study were extracted'.

Another (very) minor point relating to the abstract, I think the categories from the survey can be stated in full. These can therefore be presented as 'fairly happy' and 'very happy', rather than 'fairly and very happy'.

I should avoid commencing a section with 'hence' (under the 'mediational effects' heading).

Author Response

Response to reviewer 3 comments

Comments and Suggestions for Authors

This is a well-written paper that sets out a clear area for investigation. The methodology is cogently described and this provides a good platform for the findings. The discussion builds effectively on the findings and relates insights from this study to the literature corpus. My comments are limited to some suggestions and points of detail.

The first citation (Veenhoven, 2008) could be placed directly after the quote rather than at the end of the sentence (as a direct quote is utilised). 

Authors’ reply: We have placed the first citation directly after the quote.

It is mentioned in the methods that people of Malay and Indian ethnicity were oversampled. Does this require a degree of additional explanation? As a reader unfamiliar with the population of Singapore, I'm left unsure as to why this was necessary. Is it because people from these groups are harder to recruit to research?

Authors’ reply: In Singapore, the Malays and Indians form a minority of the population. Hence, the Malay and Indian ethnicity were oversampled to ensure a sufficient sample size for these population subgroups. We have added this information in the revised Methods accordingly.

It is clear that secondary analysis has been undertaken. However, in the discussion of the survey, in the methods section, the writing slightly lapses into a sense that this paper is based on primary research. A minor rewrite can underscore that this methodological discussion is referring to the previously collected data.

Authors’ reply: We agree with the reviewer's comment. We have mentioned that the current study was based on secondary analysis of data.

The data collection was undertaken in 2012/13 - is it worth acknowledging this point in the limitations section?

Authors’ reply: We have acknowledged this point in the limitation section.

It is a minor point, but data can be treated as plural in a scientific paper - so in the abstract under the methods heading, it can be presented as 'data from this study were extracted'.

Authors’ reply: We have revised it accordingly.

Another (very) minor point relating to the abstract, I think the categories from the survey can be stated in full. These can therefore be presented as 'fairly happy' and 'very happy', rather than 'fairly and very happy'.

Authors’ reply: We have revised it accordingly.

I should avoid commencing a section with 'hence' (under the 'mediational effects' heading).

Authors’ reply: We have removed the term 'hence'  in the abstract accordingly.